# What Can Mushroom Proteins Teach Us about Lipid Rafts?

**DOI:** 10.3390/membranes11040264

**Published:** 2021-04-06

**Authors:** Maja Grundner, Anastasija Panevska, Kristina Sepčić, Matej Skočaj

**Affiliations:** Department of Biology, Biotechnical Faculty, University of Ljubljana, Večna pot 111, 1000 Ljubljana, Slovenia; maja.grundner@bf.uni-lj.si (M.G.); anastasija.panevska@bf.uni-lj.si (A.P.); kristina.sepcic@bf.uni-lj.si (K.S.)

**Keywords:** membrane microdomains, lipid rafts, ostreolysin A, ostreolysin A6, recombinant OlyA, pleurotolysin A, pleurotolysin A2, nakanori

## Abstract

The lipid raft hypothesis emerged as a need to explain the lateral organization and behavior of lipids in the environment of biological membranes. The idea, that lipids segregate in biological membranes to form liquid-disordered and liquid-ordered states, was faced with a challenge: to show that lipid-ordered domains, enriched in sphingomyelin and cholesterol, actually exist *in vivo*. A great deal of indirect evidence and the use of lipid-binding probes supported this idea, but there was a lack of tools to demonstrate the existence of such domains in living cells. A whole new toolbox had to be invented to biochemically characterize lipid rafts and to define how they are involved in several cellular functions. A potential solution came from basic biochemical experiments in the late 1970s, showing that some mushroom extracts exert hemolytic activities. These activities were later assigned to aegerolysin-based sphingomyelin/cholesterol-specific cytolytic protein complexes. Recently, six sphingomyelin/cholesterol binding proteins from different mushrooms have been identified and have provided some insight into the nature of sphingomyelin/cholesterol-rich domains in living vertebrate cells. In this review, we dissect the accumulated knowledge and introduce the mushroom lipid raft binding proteins as molecules of choice to study the dynamics and origins of these liquid-ordered domains in mammalian cells.

## 1. Introduction

The plasma membrane of cells is a proteolipid bilayer that protects cells from their environment [1,2]. More than 10,000 different lipids and proteins have been identified in plasma membranes [3], with a high level of complexity seen for eukaryotic cells [4]. The plasma membrane was first described according to the fluid mosaic model [5], where the lipids and proteins have no restrictions on their mobility. Later, lateral membrane domains with different structures and characteristics were proposed [6], with one of the most studied of these domains known as “lipid rafts” [7,8,9,10,11,12]. These defined clusters are enriched in cholesterol (Chol), sphingolipids, and specific proteins that can diffuse within the membrane bilayer. Lipid rafts have been indicated to play important roles in endocytosis, exocytosis, membrane transport, host–pathogen interactions, cell signaling, cancer, immune response, cardiovascular diseases, and in many other cellular functions in health and disease [6,11,13,14,15,16,17,18]. 

These raft domains exist in a liquid-ordered phase (*l_o_*), and they are surrounded by lipids in a liquid-disordered (*l_d_*), or crystalline phase. The *l_o_* state means that although the molecules are highly condensed, they can physically exchange their position within the *l_o_* complex and with the molecules in the *l_d_* phase [19]. However, it is very difficult to define raft domains in the plasma membrane, as they are of nanoscale size with a life-time in the order of milliseconds [20,21]. To gain a better understanding of the raft theory, the quest for an appropriate detection technique is important. 

Soon after the raft hypothesis was proposed in 1997, diverse methodologies were used for their characterization. One of the first approaches was based on their isolation at low temperatures using detergents, as these domains in the *l_o_* phase were considered as detergent-resistant membranes. However, such detergent-resistant membranes can only exist after detergent treatment of the membrane, and therefore, it would not be correct to refer to lipid rafts as detergent-resistant membranes. Additionally, it has not been shown that all such raft domains are detergent resistant [22,23]. 

An alternative approach to demonstrate the existence of lipid rafts and to study them further, is to promote their assembly. Raft domains in mammalian plasma membranes are formed in the presence of at least 25 mol% Chol at 37 °C. Therefore, with depletion of membrane Chol using methyl-β-cyclodextrin, some information about membrane domain structure before and after raft disruption can be gained [22]. However, as the depletion of Chol can affect many processes in the cell, modifying the Chol content to study lipid rafts does not seem to be the most appropriate method [16].

Fluorescent lipid probes that specifically sense a particular membrane lipid phase and partition within it, represent an important class of tools for the visualization of raft domains in natural and artificial membranes [22]. Although their contributions to lipid raft research have been substantial, the use of fluorescent lipid probes in living cells needs to be carefully evaluated. The most common probes can be misleading; as compared to an unlabeled lipid, a fluorescently labeled lipid might partition into both the *l_o_* and *l_d_* domains, or might even partition into a different domain. Nitrobenzoxadiazole- and 4,4-difluoro-4-bora-3a,4a-diaza-s-indacene (BODIPY)-labeled sphingomyelin (SM), for example, has been shown to partition into the *l_d_* phase, although natural SM partitions into the *l_o_* phase [24]. 

The lipid probes that are known as environment-sensitive dyes (e.g., molecular rotors, solvatochromic and electrochromic dyes) can partition into both lipid phases, and as they react to the properties of the individual phase, they can allow for direct recognition of these phases by fluorescence ratiometric imaging. However, these dyes have not been well explored to date [25]. Alternatively, lipid-specific/sensing molecules can be used, such as the polyene macrolide antibiotic filipin, which interacts specifically with 3β-OH sterols and, thus, can be used as a Chol marker. However, its use has some disadvantages, as it disrupts lipid rafts and has poor fluorescence properties [26]. Using hydrophobic probes for staining lipids in natural membranes can also be difficult in terms of interpretation. Due to their high migration properties, these probes rapidly stain all intracellular lipid compartments, so they are not the most appropriate for live-cell imaging [27]. In this regard, proteins that target specific raft-enriched molecules might represent a better option [27].

For many years, cholera toxin B subunit (CT-B) has been widely used for detecting lipid rafts [28,29,30], and it has provided important insights into membrane trafficking. CT-B specifically binds to the raft-residing ganglioside GM1. Once internalized, the cholera toxin moves to the *trans* Golgi network and then to the endoplasmic reticulum. It is either recycled between the Golgi complex and the endoplasmic reticulum or passes into the cytosol, where its A subunit can induce toxicity [31]. However, the use of CT-B for lipid raft detection also has some disadvantages. These include the nonspecific binding of CT-B to glycans at the cell surface and the induction of local rearrangements of cell membranes after binding to the GM1 receptor, with the consequent formation of lipid domains [32,33]. Finally, some mammalian cells are depleted in, or have undetectable amounts of GM1, so they cannot be probed by CT-B [27,34,35]. 

Recently, nontoxic mushroom proteins and their fluorescently labeled derivatives have become a tool of choice for the visualization of SM/Chol-rich raft domains in living and fixed mammalian cells. These interesting mushroom proteins are relatively small (15–22 kDa) and acidic (pI, 4.8–6.2), and have also been proposed to be useful in many biotechnological applications (Table 1 and Table 2). The SM/Chol-binding proteins from the genus *Pleurotus* (aegerolysins) act as cytolytic proteins when in concert with a partner protein with a membrane attack complex-perforin (MACPF) domain. However, this is not the case for nakanori, which is a lipid raft sensing protein from the mushroom *Grifola frondosa* (Table 1). These proteins were the first—and to date, they remain the only—molecules that simultaneously bind only to mixtures of the most abundant raft lipids, SM and Chol, and not to individual molecules of these two lipid species [28,36,37,38,39,40]. Their specific binding to SM/Chol-rich membrane domains was also demonstrated using in vitro cell models with co-localization studies with endogenous raftophylic proteins (Table 2). As with every raft probe mentioned so far, raft sensing proteins from mushrooms also have some disadvantages. As discussed further in the text, aegerolysin-based probes might induce the unwanted vesiculation of cells if used in exaggerated concentrations, as discussed in Section 2.2. 

Amino acid sequence alignment of *Pleurotus* aegerolysins is shown in Figure 1A, where the differences in amino acid compositions between these proteins and nakanori can be clearly seen. The differences between these two types of SM/Chol-sensing proteins, *Pleurotus* aegerolysins and nakanori, are observed also at higher levels of protein structure, where aegerolysins are mainly β-structured, while nakanori is mainly composed of α-helices (Figure 1B). Below, we dissect out the accumulated knowledge from recent years, and we introduce the mushroom SM/Chol-binding/sensing proteins ostreolysin A (OlyA) and ostreolysin A6 (OlyA6), recombinant (r)OlyA, pleurotolysin A (PlyA), pleurotolysin A2 (PlyA2), and nakanori as probes for studying lipid rafts in living cells.

## 2. Sphingomyelin/Cholesterol Binding Proteins

### 2.1. Ostreolysin A

In 2002, a 15 kDa protein ostreolysin (OlyA, also Oly) was purified from the edible mushroom *Pleurotus ostreatus* [51]. OlyA belongs to the aegerolysin family of proteins that are found in several mushrooms, molds, plants, and bacteria [41,52]. Early experiments with isolated native OlyA showed that it has a low isoelectric point, and it is thermolabile and cytotoxic to erythrocytes and several cell lines [36,53,54,55]. It is expressed during the formation of primordia (an organ or tissue in its earliest recognizable stage of development) and fruiting bodies (a macroscopic reproductive structure produced by some fungi) of the oyster mushroom, and therefore, it was believed to have a role in mushroom fruiting [51,53]. In 2004, it was discovered that OlyA recognizes and permeabilizes membranes rich in Chol (>30 mol%) and SM [36], and that binding is abolished after methyl-β-cyclodextrin pre-treatment of the cells [45]. However, it was later discovered that this native isolate was contaminated with traces of another protein, pleurotolysin B (PlyB) [42], which accounted for the observed cytolytic effects, as described in the next section.

### 2.2. Ostreolysin A6

In 2013, it was discovered that the genome of *P. ostreatus* has several genes that encode different aegerolysins. One of these was expressed and the corresponding aegerolysin showed 78% amino acid identity with OlyA (Figure 1), and was named as ostreolysin A6 (OlyA6). OlyA6 was expressed in a recombinant form and was shown to be nonlytic by itself. Sedimentation assays and surface plasmon resonance revealed that OlyA6 is responsible for the recognition of and interaction with SM/Chol-rich artificial lipid membranes and with natural membranes with SM/Chol-rich microdomains. Recently, it was confirmed that OlyA6 binds specifically to SM/Chol complexes, and not to free SM, and that this specificity is controlled by a single glutamic acid residue near the binding pocket [56]. Moreover, it appears that OlyA6 traps and stabilizes these complexes, which allows direct measurements of their levels. 

As its lipid-binding characteristics indicated that OlyA6 can be used as a new raft marker, Skočaj et al. produced a fluorescent fusion protein OlyA–mCherry [37,42]. Surface plasmon resonance revealed that OlyA6–mCherry has essentially the same binding characteristics as nonlabelled recombinant OlyA6. Labeling of SM/Chol domains in the plasma membrane of live and fixed MDCK (Madin-Darby Canine Kidney) cells with 1 µM OlyA6–mCherry did not induce cell lysis. As for the isolated native OlyA [45], this binding was abolished by methyl-β-cyclodextrin or sphingomyelinase pretreatment. Interestingly, the labeling of MDCK cells with OlyA6–mCherry induced formation of vesicles of 2 µm to 10 µm in diameter when longer incubation times or higher protein concentrations were used [46]. The same was seen in studies with artificial large unilamellar vesicles composed of extracted total erythrocyte lipids or SM/Chol in a 1:1 molar ratio [42], and with Chinese hamster ovary cells [57], neuroblastoma NG108-15 cells [58] and different types of blood cells [46]. This vesiculation was not surprising, as similar observations had been reported for the Chol-binding toxin streptolysin O, and were explained as a physical effect of the protein binding to the membrane and not as a result of apoptosis [59]. Alternatively, clustering of OlyA6 molecules might induce locally imposed membrane curvature that can lead to vesicle formation. These vesicles induced by OlyA6–mCherry binding contained several cytoplasmic proteins and were proposed to be used for sampling of the cytosolic content, or as model plasma membrane vesicles [46].

Co-localization studies showed that CT-B labels different domains compared to OlyA6, which confirmed that raft domains are highly heterogeneous (Figure 2). Furthermore, when OlyA6–mCherry was expressed in MDCK cells, it did not bind to intracellular membranes. The reason for this might be the lower content of SM in membranes exposed to cytosol, as needed for OlyA6–mCherry binding. OlyA6–mCherry was also used to study caveolin-dependent internalization pathways of raft membranes, as it internalizes *via* caveolae, and can be detected near the Golgi complex after 90 min [37]. CT-B, in contrast, follows a different pathway to reach the Golgi complex and endoplasmic reticulum [60]. These internalization studies led to the conclusion that OlyA6–mCherry can be used as a tool for delivering pharmacologically active substances to specific intracellular compartments [37].

Furthermore, it was shown that OlyA6 acts in concert with PlyB, which is also produced by *P. ostreatus*, as a 59 kDa protein that has an MACPF domain. OlyA6 and PlyB form a cytolytic complex, where OlyA6 serves as the binding component and recruits PlyB to the membrane surface, which leads to the formation of 13-meric transmembrane pores [42,61]. In combination with PlyB, OlyA6 can be used to selectively eliminate cells with more SM/Chol domains; e.g., in some cancer cell types [47]. OlyA6/PlyB complexes have been demonstrated to be good candidates in the search for a cure for bladder cancer. Pretreatment of low-grade human papillary cancer cells and high-grade invasive human urothelial cancer cells with high concentrations of OlyA6/PlyB decreased cell viability and resulted in cell necrosis. Compared to nontransformed normal urothelial cells, the Chol concentrations in both of these transformed cell lines were higher, and therefore, the cytotoxic effects were only seen in the cancer cells. Metastatic urothelial carcinomas can be resistant to chemotherapy, and therefore, the OlyA6/PlyB complex has become a very interesting tool for a potential cure for this type of cancer [47].

Recently, it was revealed that OlyA6 has another high affinity lipid receptor, ceramide phosphoethanolamine (CPE), which is present at very low levels in mammals, but is the main sphingolipid in invertebrate cell membranes [62]. OlyA6 shows 1000-fold stronger binding to CPE/Chol lipid vesicles than to SM/Chol [63]. Moreover, OlyA6 binds strongly to equimolar 1-palmitoyl-2-oleoyl-glycero-3-phosphocholine/Chol vesicles supplemented with less than 5 mol% CPE [64]. This discovery led to the consideration that OlyA6 can also be used as a CPE marker in insect cells and tissues [63,64,65]. Additionally, the permeabilization effects of OlyA6/PlyB complexes have been shown for artificial lipid systems with CPE, and Sf9 cells and larvae of selected insect pests. OlyA6/PlyB complexes (molar ratio, 12.5:1) permeabilize lipid vesicles with <5 mol% CPE and are toxic to Sf9 cells, and to Western corn rootworm and Colorado potato beetle larvae [64]. Thus, OlyA6/PlyB complexes have been proposed as new environmentally friendly bioinsecticides [64,66]. Furthermore, it is highly likely that mushrooms use aegerolysin proteins together with their MACPF-protein partners as defense mechanisms against environmental pests in their natural habitats.

Taken together, recombinant OlyA6 is an important tool to study SM/Chol domains in mammalian cells [37,56,57]. It is a small and stable protein, and it is not toxic for vertebrate cells [37,64]. OlyA6 has revealed a heterogenic pool of membrane domains, and it has been shown to be appropriate for studying the turnover of lipid rafts in living mammalian cells [37].

### 2.3. rOlyA

Oyster mushrooms are recognized as having high medicinal value [67]. Recently, a new aegerolysin, produced by *P. ostreatus* (in recombinant form), was shown to have anti-cancer activity, and can also be used to fight obesity. This aegerolysin was first produced in 2017, and it was named recombinant ostreolysin (rOly) [43]. rOly is almost identical to OylA6 (Figure 1), but it has one amino acid substitution (valine to isoleucine) at position 50. rOly has not been used as a lipid raft marker, but it was suggested that its SM/Chol binding characteristics are responsible for its anti-proliferative, pro-apoptotic effects against colon cancer cells, and for the treatment of metabolic disorders, as summarized in the following paragraphs [43,48,49].

Recombinant Oly induces apoptosis of human and mouse colon cancer cells in vitro, as these cells are enriched in lipid rafts. After endocytosis, membrane-bound rOly promotes programmed cell death in colon cancer cells in a dose-dependent manner, which was not significant in a normal intestinal cell line. After internalization, rOlyA co-localizes with β-III tubulin, and it was suggested that this interaction, which affects microtubule dynamics, leads to apoptosis. The anticancer potential of rOlyA was also confirmed in vivo in mouse models, where rOly reduced tumor growth by modulation of signal-transduction pathways, and thus, effects on inflammatory processes and decreased angiogenesis [43]. In contrast to OlyA6, which shows anticancer activity against bladder cancer cells in vitro when applied in combination with PlyB, rOlyA has antitumor activity without its MACPF protein partner. 

Recombinant Oly has also been shown to be an interesting candidate for combating obesity and other metabolic disorders. There are two types of adipocytes in mammals (i.e., white and brown). The white adipocytes serve for storage of energy in the form of fat. Increased numbers of these cells can lead to obesity or to nonalcoholic fatty liver disease if stored in the liver, which can result in insulin insensitivity. Brown adipocytes contain more mitochondria, which can produce more heat and less ATP as a result of uncoupling protein activity [48,49]. Higher numbers of brown adipocytes are, therefore, favorable against obesity. White preadipocytes are enriched in lipid rafts, which correlates with their sensitivity to aegerolysin treatment [68]. Recombinant Oly induces “browning” of adipocytes in vitro, as shown by increased brown-adipocyte-specific markers. As a result, the brown fat-like phenotype of in vitro adipocytes increases the activation of mitochondria and mitochondrial biogenesis, including the expression of thermogenic markers and inhibition of inflammation. In vivo studies have demonstrated that rOly can dramatically reverse accumulation of fat in the liver, suppress *de novo* lipogenesis, and reduce glycolysis and hepatic inflammation. As a result, serum levels of free fatty acids in mice on a high fat diet treated with rOly were lower than in untreated mice. rOly has, therefore, been proposed as an appropriate candidate for treatment of metabolic disorders, such as obesity, hyperlipidemia, and nonalcoholic fatty liver disease. 

### 2.4. Pleurotolysin A 

Then, in 1979, the 15 kDa protein pleurotolysin (Ply) was isolated from *P. ostreatus* and was shown to have hemolytic activity towards mammalian erythrocytes. Initial investigations with erythrocytes from various animals showed the importance of SM for Ply binding [69]. In 2004, a native cytolysin was isolated, and as in the case of Oly, it was shown to be composed of A and B subunits. The A subunit (PlyA) is a 15 kDa aegerolysin that binds to membranes, and in combination with the 59 kDa MACPF-domain-containing component B (PlyB), it forms transmembrane pores [50]. The primary structure of PlyA is 79% and 94% identical to OlyA and OlyA6, respectively (Figure 1). While studies of OlyA6 were more cell-labeling oriented, PlyA studies have focused on the structure and mechanism of PlyA binding to target membranes. The crystal structure of PlyA was obtained by a combination of X-ray diffraction and cryo-electron microscopy, and this showed that PlyA has a β-sandwich structure similar to the actinoporin family of pore-forming toxins [70]. The N-terminal region, which is responsible for pore formation in actinoporins [71], is absent in PlyA; therefore, similarly to OlyA6, PlyA cannot form transmembrane pores by itself [50]. 

PlyA binds specifically to SM/Chol-rich membranes, which was demonstrated for liposomes with different compositions. The binding was detected only with SM/Chol liposomes and not where SM was replaced with glycerophospholipid species [50]. One explanation of the importance of SM might lie in the PlyA structure; there are a few negatively charged amino acids in a region that is similar to a region in Asp-hemolysin, the first of the isolated aegerolysin-like protein from the filamentous fungus *Aspergillus fumigatus* [44,72]. These residues were proposed to be responsible for identification of choline. Analysis of the secondary structure of PlyA also indicated that the N-terminal part of 14 amino acids can interact with the hydrophobic part of the membranes through its β-sheet [44]. This could be linked to the same binding pocket present in both OlyA6 and PlyA, which is responsible for recognizing the specific conformation of SM in SM/Chol complexes. Thus, PlyA can also distinguish between SM conformations and has very similar binding properties to OlyA6. For PlyA binding, the Chol content should be ≥30 mol% Chol. As for OlyA6, PlyA can form pores in artificial membranes and cell membranes when PlyB is present [50]. Two PlyA monomers bind to the membrane to form a V-shaped dimer, which is then recognized by PlyB. After the conformational change of PlyB and insertion of β-hairpins through the membrane, the 13-fold pore is formed with the membrane-inserted β-barrel [70]. It has been suggested that the optimal molar PlyA:PlyB ratio for pore formation is 3:1. 

### 2.5. Pleurotolysin A2

Sedimentation assays with an extract from the mushroom *P. eryngii* gave rise to identification of another protein that bound to SM/Chol liposomes. This protein was shown to be a PlyA ortholog by Bhat et al. [38], so they named it pleurotolysin A2 (PlyA2). Similar to OlyA6 and PlyA, PlyA2 belongs to the aegerolysin protein family [38] and was shown to be a 15 kDa polypeptide. Compared to PlyA, PlyA2 has two amino acid substitutions [73], while it shares 93% amino acid sequence identity with OlyA6 [38].

As PlyA2 specifically recognizes SM/Chol-rich membrane domains, it also became interesting as a potential lipid raft marker. The binding properties of recombinant PlyA2 fused with EGFP (Enhanced Green Fluorescent Protein) at the C-terminus (PlyA2-EGFP) were tested on artificial and natural membranes. PlyA2-EGFP bound to liposomes composed of SM/Chol (1:1 molar ratio), and this binding was abolished if Chol or SM were excluded from the liposomes, which indicated the importance of the presence of both SM and Chol [38]. As in the case of OlyA [74], alterations in the A and B rings of the sterol appear to be important for PlyA2 binding, whereas modifications of the sterol hydrocarbon side-chain do not influence the binding. The binding of PlyA2 is, therefore, mostly restricted to the head-group of the complex of SM and sterol [38].

In addition, the binding and toxicity properties of PlyA2 and PlyA2-EGFP were examined with natural membranes. When different concentrations of these proteins were incubated for 30 min with sheep erythrocytes, no hemolytic effects were seen [38]. As in the case of OlyA6–mCherry in living MDCK cells [37,64], incubation of Hela cells with PlyA2-EGFP [38] showed successful labeling, with no toxicity. Similarly, the binding of PlyA2-EGFP to HeLa cells was abolished after pretreatment with sphingomyelinase or methyl-β-cyclodextrin. This indicated that PlyA2-EGFP is a suitable marker of SM/Chol-rich membrane domains. Co-localization studies were performed to analyze the surface distribution of PlyA2-EGFP compared to other raft markers (Figure 3). PlyA2-EGFP showed 45% co-localization with the raft-associated glycosylphosphatidylinositol-anchored protein CD59. After permeabilization by freezing and thawing (to avoid lipid loss), the EGFP signal was detected in late endosomes [38]. Similar to OlyA6–mCherry [37], PlyA2-EGFP did not label the cytoplasmic leaflet of the cells [38].

However, PlyA-EGFP was later shown to have another lipid receptor with higher affinity, as reported for OlyA6. PlyA-EGFP binds to CPE/Chol membranes and can also recognize and bind to Chol-free CPE. PlyA2-EGFP successfully labeled CPE in Kc167 embryonic *Drosophila* cells, and in the larvae of the same insect [63]. Panevska et al. [64] suggested that PlyA2/PlyB complexes can be used as new bioinsecticides, as discussed for OlyA6/PlyB complexes. 

### 2.6. Nakanori

The most recent SM/Chol binding protein is nakanori (“mid-raft rider” in Japanese). It was isolated from the edible mushroom *Grifola frondosa* in 2017. In contrast to OlyA, OlyA6, rOlyA, PlyA, and PlyA2, it does not belong to the aegerolysin protein family. The globular structure of this protein is different from the aegerolysin fold. Nakanori is composed of antiparallel β-strands, and it has two helical regions. Topologically, nakanori shows similarities to the actinoporin protein family, although these proteins share only 17% sequence identity. The additional 30 residues at the N-terminus and the lack of amphipathic character of helix 3 might be responsible for the nonlytic character of this protein [40].

The 23 kDa nakanori protein binds strongly to SM/Chol liposomes with >20 mol% Chol, and not to other phospholipid/Chol mixtures. If Chol is substituted with ergosterol, binding of nakanori can still be detected, but the membrane binding is abolished with epicholesterol and cholesteryl acetate. This suggests that the 3β-hydroxyl function is important for nakanori binding. To clarify whether nakanori binds to pre-existing SM/Chol complexes or induces complex formation, sedimentation assays were performed with ternary mixtures of SM/1,2-dipalmitoyl-*sn*-glycero-3-phosphocholine/Chol and SM/1-palmitoyl-2-oleoyl-glycero-3-phosphocholine/Chol. As the increasing content of 1,2-dipalmitoyl-*sn*-glycero-3-phosphocholine markedly reduced the binding of nakanori, and the presence of 1-palmitoyl-2-oleoyl-*sn*-glycero-3-phosphocholine reduced the binding only moderately (which is consistent with the preferred association of Chol with SM over 1-palmitoyl-2-oleoyl-*sn*-glycero-3-phosphocholine), the conclusion was that nakanori does not induce the formation of lipid domains by itself. Measurements with a quartz crystal microbalance with dissipation monitoring (QCM-D) for binding affinity defined the dissociation constant *K_D_* of recombinant nakanori to liposomes composed of SM/Chol (1:1, molar ratio) [40], which is stronger than those for OlyA6 and PlyA2 [38] (Table 2).

Fluorescently labeled nakanori (nakanori-EGFP) was not toxic for HeLa cells, and treatment with sphingomyelinase and methyl-β-cyclodextrin abolish the binding of both labeled and nonlabeled nakanori. The intracellular labeling of HeLa cells showed that nakanori co-localizes with the SM-binding protein lysenin, and also with *bis*(monoacylglycero)phosphate/ lysobisphosphatidic acid, CD63 and Lamp1, which are enriched in the internal membranes of late endosomes [75,76] (Figure 4). Nakanori did not co-localize with markers of early endosomes, the Golgi complex, or the endoplasmic reticulum [40].

Nakanori has been shown to be a useful tool for the detection of alterations in membrane dynamics in some diseases [40]. The genetic disease known as Niemann–Pick type C (NPC) is defined by intracellular accumulation of free Chol in late endosomes of fibroblasts [75]. In cultured skin fibroblasts from patients with NPC, the detailed lipid composition, structure and dynamics of cell surface lipid rafts were investigated by Makino et al. [40] using nakanori. Fluorescently labeled nakanori showed different binding properties for normal and NPC cells. In normal skin fibroblasts nakanori bound homogeneously, in contrast to NPC skin fibroblasts, where the staining was heterogeneous and the average radius of the labeled clusters was larger. Nakanori revealed micro-scale segregation of lipid domains in these NPC cells. Additionally, the mobility of the fluorescently labeled nakanori was lower in NPC cells compared to normal cells. Only small amounts of nakanori were internalized with these human skin fibroblasts. These results showed that nakanori can be used to detect differences in lipid raft organization in different cells [40]. An interesting application of nakanori has been proposed to affect the life cycle of the influenza virus. During the later stages or during the entire time of an infection, the addition of higher concentrations of nakanori inhibited the release of the virus from MDCK cells. This indicates the importance of specific membrane domains and targeting of these domains in the development of anti-viral drugs [40].

Considering all of the data presented here, we can conclude that nakanori can be used to detect SM/Chol domains, in addition to altered distributions of lipid domains in some diseases. It has also become interesting as a potential probe for targeting lipid domains in anti-viral treatments.

## 3. Conclusions

The lipid raft hypothesis met some challenges at first, but now it appears to be largely accepted and supported, although some questions still remain unresolved [3,12,77]. Lipid rafts have important roles in many pathologies, including: Alzheimer’s disease; Parkinson’s disease; cardiovascular, prion and immune diseases; in addition to HIV. Therefore, lipid raft domains have become an interesting target in the search for a prevention and cure for such diseases [4,11,18]. 

Lipid raft domains cannot be easily detected as they have a size below optical resolution and they are heterogeneous and have a short life-time [20,21]. Many protein-based probes have been developed to visualize lipid rafts [20,78,79], but which one to use requires careful consideration. Some probes, such as CT-B, can induce rearrangements of membrane domains after binding, and/or clustering of different raft components [80]. Nonlytic fluorescently labeled proteins from some mushrooms might represent suitable tools for lipid raft domain visualization in living and fixed cells. OlyA, OlyA6, rOlyA, PlyA from *P. ostreatus*, PlyA2 from *P. eryngii*, and nakanori from *G. frondosa* have high specificities for mammalian cell membranes, as they bind to SM/Chol-rich domains. As the requirements for SM or Chol content differ among these proteins [37,38,40,46,50,78], there is a choice to be made in terms of which protein to use for labeling specific cells or cell membranes. As well as binding to SM/Chol, OlyA6 and PlyA2 bind even more strongly to CPE/Chol domains. Therefore, these can be used as probes for lipid domains in invertebrates [63,64,81]. In combination with an MACPF protein, PlyB, OlyA6, PlyA, and PlyA2 form pores in lipid membranes, and can, therefore, be used for treating some cancers [47]. Additionally, due to their SM/Chol binding properties, these proteins are promising agents for anti-viral treatments [40], and potentially could be used also against COVID-19 infection. They can also be used for detecting SM/Chol-related metabolic disorders [49] and as potential anti-cancer treatments through their pro-apoptotic activities, with targeting seen for colon cancer cells [43]. In addition, the use of these aegerolysins might reveal the mechanisms of regulation of lipid homeostasis [56], which is crucial for the modulation of many cell signalling processes [6,82].

As the kingdom of fungi consists of more than a million different species, we also suspect that they contain an arsenal of diverse lipid-binding proteins that are yet to be defined, among which, many might show high affinity for specific membrane domains. Mushroom proteins that label SM/Chol-rich membrane domains have helped us to demonstrate that lipid rafts actually exist in living systems, and they have provided us with a useful, and indeed indispensable, toolbox for studying these domains in living vertebrate cells.

## Figures and Tables

**Figure 1 membranes-11-00264-f001:**
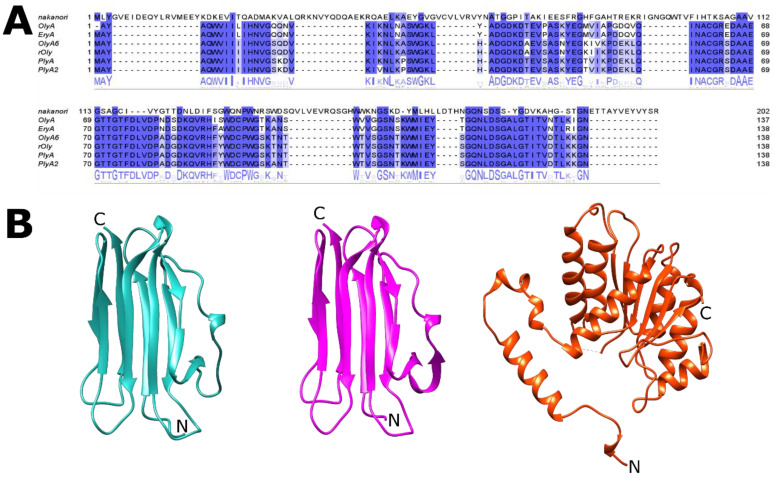
**Fungal lipid raft binding proteins.** (**A**) Jalview representations of amino acid sequence alignment of the aegerolysin proteins: ostreolysin A (OlyA; NCBI AAX21097.1); ostreolysin A6 (OlyA6; UniProtKB/Swiss-Prot: P83467.2); recombinant ostreolysin (rOly; NCBI KDQ25828.1); pleurotolysin A (PlyA; NCBI BAD66666) from *Pleurotus ostreatus*; pleurotolysin A2 (PlyA2; NCBI BAN83906.1) from *Pleurotus eryngii*, and nakanori (NCBI BAO31550.1) from *Griofola fondosa*. Positions of the highest similarity are shaded in blue. (**B**) Representative aegerolysin crystal structures of OlyA6 (pink; PDB 6MYI), PlyA (green; PDB 4OEB) and nakanori (orange; PDB 5H0Q), visualised using the UCSF (University of Californiaa, San Francisco). Chimera package. The N-termini and C-termini are indicated.

**Figure 2 membranes-11-00264-f002:**
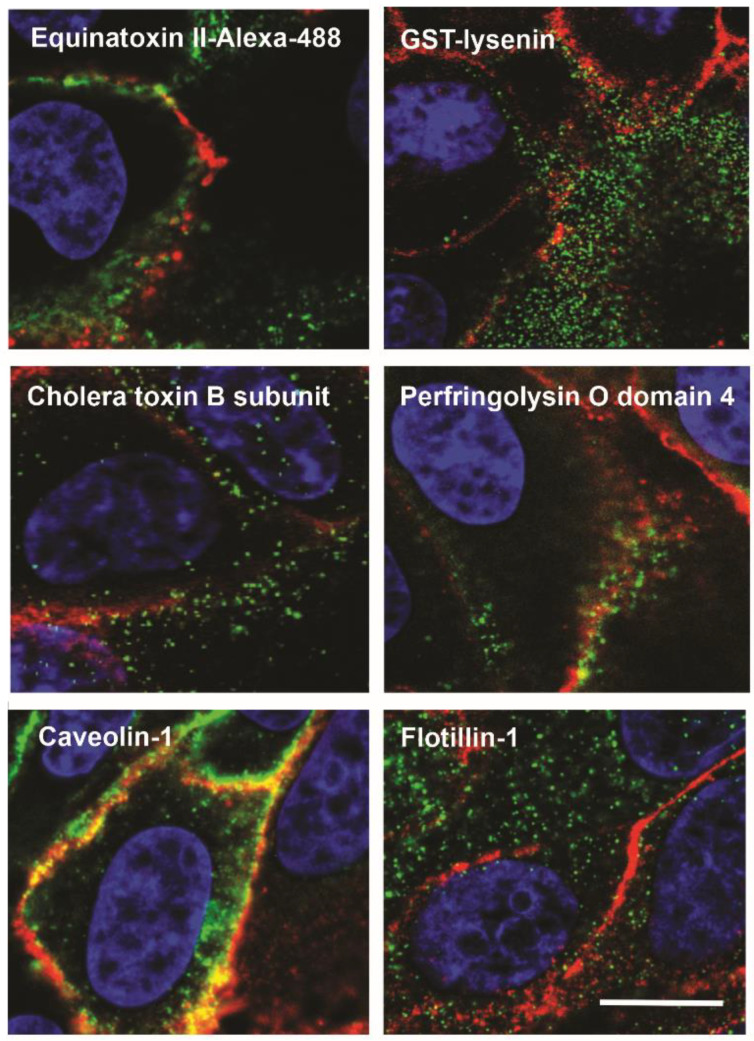
**Distribution of OlyA–mCherry and established lipid raft markers.** Representative fluorescent images of double immunolabeling of fixed MDCK cells treated simultaneously for 10 min with OlyA–mCherry (1 μM; red) and toxin-derived probes (green) that detect sphingomyelin (lysenin, equinatoxin II), cholesterol (domain 4 of perfringolysin O), ganglioside GM1 (cholera toxin, subunit B), or intrinsic proposed membrane markers (caveolin-1, flotillin-1). Blue, DAPI (diamidino-2-phenylindole). Scale bar: 20 µm.

**Figure 3 membranes-11-00264-f003:**
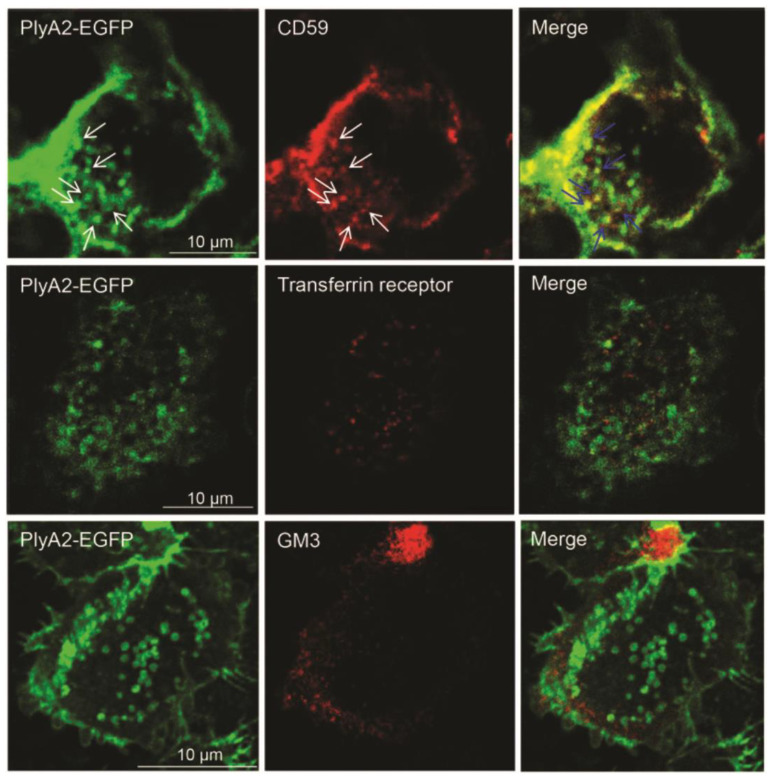
**Cell surface distribution of PlyA2-EGFP (green) and lipid raft markers (red).** Fixed HeLa cells were doubly immunolabeled with PlyA2-EGFP and antibodies against raft-associated glycosylphosphatidylinositol-anchored protein CD59 and ganglioside GM3. Alternatively, cells were labelled with non-raft transferrin followed by fixation and PlyA2-EGFP labelling. Arrows, partial co-localization between PlyA2-EGFP and CD59. Based on dot-by-dot comparisons, about 45% of CD59-positive dots co-localized with PlyA2-EGFP. In contrast, only 5% of the non-raft transferrin receptor was visualized with plasma-membrane-bound Alexa-labelled transferrin. Ganglioside GM3 did not show significant co-localization with SM/Chol domains labeled by PlyA2-EGFP. Adapted from [38].

**Figure 4 membranes-11-00264-f004:**
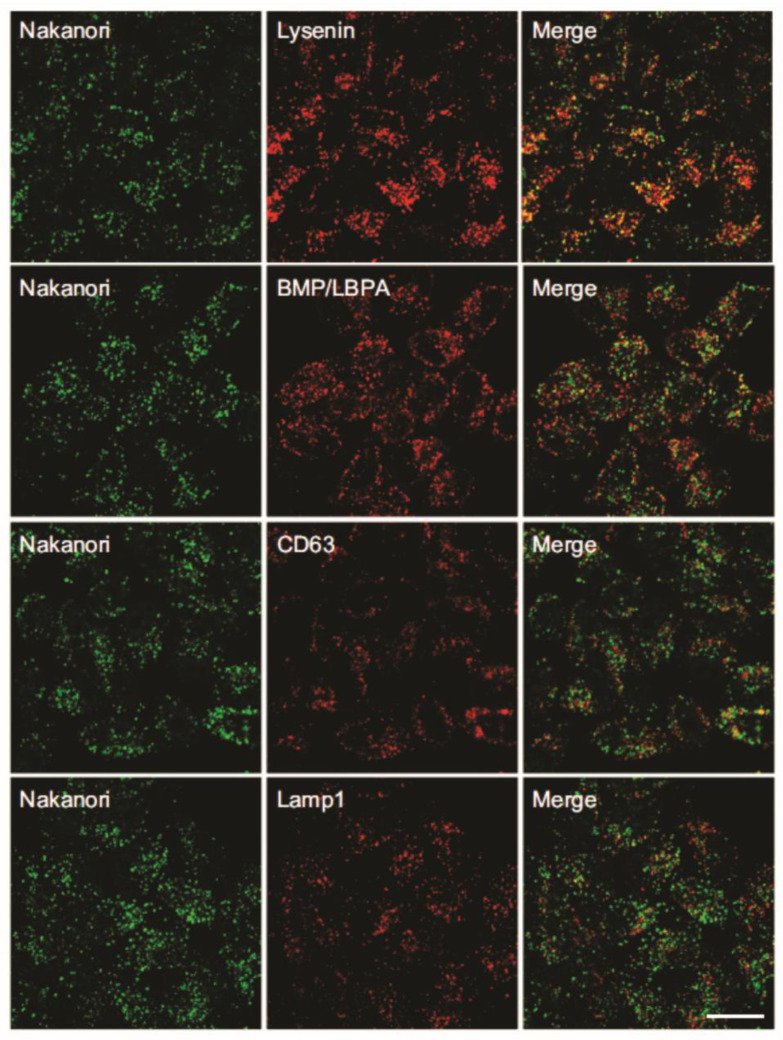
**Intracellular distribution of nakanori and organelle markers.** Representative fluorescent images of double immunolabeling of Hela cells with nakanori (green) and organelle markers (red) showing co-localization of nakanori with lysenin, and with markers of the internal membranes of late endosomes, BMP/LBPA (*bis*(monoacylglycero)phosphate/lysobisphosphatidic acid) and CD63. Some co-localization of nakanori was seen with Lamp 1, as a marker of the limiting membrane of late endosomes. Scale bar: 20 μm. Adapted from [40].

**Table 1 membranes-11-00264-t001:** Basic biochemical properties of the mushroom lipid raft binding proteins.

Protein	NCBI Accession Code	Source Organism	MW ^a^ (kDa)	pI ^b^	Cytolytic Protein Partner	Reference
OlyA ^c^	AAX21097.1	*P. ostreatus*	14.9	4.8	PlyB ^h^	[41]
OlyA6 ^d^	AGH25589.1	*P. ostreatus*	15.0	5.5	Ply B	[37,42]
rOly ^e^	KDQ25828.1	*P. ostreatus*	15.0	5.5	Ply B	[43]
PlyA ^f^	BAD66668.1	*P. ostreatus*	14.9	5.9	Ply B	[44]
PlyA2 ^g^	BAN83906.1	*P. eryngii*	15.0	5.5	Erylysin B	[38]
Nakanori	BAO31550.1	*G. frondosa*	22.6	6.2	/	[40]

^a^ molecular weight; ^b^ isoelectric point; ^c^ ostreolysin A; ^d^ ostreolysin A6; ^e^ recombinant Oly; ^f^ pleurotolysin A; ^g^ pleurotolysin A2; ^h^ pleurotolysin B.

**Table 2 membranes-11-00264-t002:** The main characteristics of the mushroom lipid raft binding proteins.

Protein	*K_D_* (nM; SM/Chol)	Co-Localization	Cell Model	Biotechnological Applications	Reference
OlyA	>1000	/		Lipid raft biomarker	[36,41,45]
OlyA6	ND	Caveolin 1	MDCK cells	Lipid raft biomarker; OlyA6-induced vesicles for sampling of cytoplasm or as model plasma membrane vesicles; tool for delivering pharmacologically active substances to specific intracellular compartments; treatment of bladder cancer (with PlyB)	[37,42,46,47]
rOly	ND	ND	ND	Treatment of colorectal tumors and metabolic disorders	[43,48,49]
PlyA	ND	ND	ND	Lipid raft biomarker	[50]
PlyA2	>1000	CD59	Hela cells	Lipid raft biomarker	[38]
Nakanori	141	Lysenin; BMP/LBPA; CD63; Lamp1	Hela cells	Lipid raft biomarker; tool for detecting alterations in membrane dynamics (detection of Niemann–Pick disease); inhibition of release of influenza virus from MDCK cells	[40]

*K_D_* dissociation constant; MDCK, Madin-Darby Canine Kidney (cells); ND, not determined.

## Data Availability

Not relevant.

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
