# Peer review of "What Can Mushroom Proteins Teach Us about Lipid Rafts?"

_membranes, 2021, doi:10.3390/membranes11040264_

Round 1
Reviewer 1 Report
The review "What can mushrooms teach us about lipid rafts?” is well written and organised. The review is timely, relevant and fits in the scope of the journal. The review provides all background to the reader less familiar with the subject, thereby including all necessary international state-of-the-art.
I have only one comment regarding the division of chapters. In my opinion, it would be clearer to break down the parts as follows:
1.Introduction
2. "target proteins"
2.1. Ostreolysin A
2.2. Ostreo....
2.3. ...
3. Conclusions
Author Response
Authors' replies to Reviewers’ and Editor’s comments:
Reviewer #1
The review "What can mushrooms teach us about lipid rafts?” is well written and organised. The review is timely, relevant and fits in the scope of the journal. The review provides all background to the reader less familiar with the subject, thereby including all necessary international state-of-the-art. I have only one comment regarding the division of chapters. In my opinion, it would be clearer to break down the parts as follows:
1.Introduction
- "target proteins"
2.1. Ostreolysin A
2.2. Ostreo....
2.3. ...
- Conclusions
Response from the Authors:
Thank you for the suggestion. The chapters were modified as Reviewer #1 suggested, and the corrections are marked in yellow.

Reviewer 2 Report
This is a review paper summarizing information on lipid raft binding proteins from mushrooms. These proteins are a relatively new and underappreciated tool to study membrane structure and I am not aware of other in-depth reviews dedicated specifically to these proteins. This is a very useful resource that I am sure it will be widely used. It is up-to-date, comprehensive and well written. However, it is somewhat biased: it would be beneficial if the authors took a more critical approach discussing not only advantages, but also known problems with using these proteins. One important issue that definitely needs to be discussed is that often there is limited overlap between staining with these proteins and other established lipid raft markers bringing some uncertainty to what they actually bind.
Specific comments:
- I would recommend to have another look at the title. The current title is catchy, but an immediate interpretation (e.g. when scanning PubMed) is that this is about lipid rafts in mushrooms. By the way, are there lipid rafts in the mushrooms?
- It would be good if the authors could briefly discuss what is the natural function of these proteins in mushrooms.
- 2: I do not quite agree with reasoning behind criticism of older methods to investigate lipid rafts. At times, this reasoning simply represents Heisenberg’s Uncertainty Principle (to investigate the object you need to disturb it – apply detergents, or insert a fluorescent probe etc), which applies to any method and is not a valid point of criticism of any particular method. The methods mentioned do have many limitations, it’s just not this.
- Lines 170-172: a straight forward explanation is that OlyA6-mCherry does not bind to lipid rafts, but to a different domain, isn’t it? This needs to be discussed.
- Minor: my understanding is that the target audience of this review are experts in membranes, not experts in mushrooms. I would recommend including some brief explanations of botanical terms, it will make reading stumble-free.
Author Response
Reviewer #2
This is a review paper summarizing information on lipid raft binding proteins from mushrooms. These proteins are a relatively new and underappreciated tool to study membrane structure and I am not aware of other in-depth reviews dedicated specifically to these proteins. This is a very useful resource that I am sure it will be widely used. It is up-to-date, comprehensive and well written. However, it is somewhat biased: it would be beneficial if the authors took a more critical approach discussing not only advantages, but also known problems with using these proteins. One important issue that definitely needs to be discussed is that often there is limited overlap between staining with these proteins and other established lipid raft markers bringing some uncertainty to what they actually bind.
Response from the Authors:
We totally accept the criticism of the Reviewer #2, especially regarding the disadvantages of using these proteins for staining lipid rafts. The answers regarding these questions are summed up in Queries #2 and #3.
Query #1:
I would recommend to have another look at the title. The current title is catchy, but an immediate interpretation (e.g. when scanning PubMed) is that this is about lipid rafts in mushrooms. By the way, are there lipid rafts in the mushrooms?
Response from the Authors:
The Reviewer #1 clearly noticed that the sphingomyelin/cholesterol binding proteins from mushrooms are relatively new and underappreciated tools to study membrane structure. Since we have been studying these proteins for the last 20 years, we aimed to use this intriguing title to invite the readers of this special issue of Membranes for further reading, and to introduce these neglected proteins as valuable tools for studding these enigmatic microdomains called lipid rafts. We feel that the lipid raft experts are aware that lipid rafts (sphingomyelin/cholesterol rich membrane domains) exist only in vertebrate cells. Although the counterparts of these particular membrane domains can also exist in other domains of life, they do not have this specific lipid composition. For example, fungi do not synthesize sphingomyelin and cholesterol, but their membrane inositol phosphosphingolipids may function as orthologs of mammalian sphingomyelin, and can form raft-like domains with ergosterol, the main fungal sterol (Mollinedo (2012) Front Oncol, 10;2:140, doi: 10.3389/fonc.2012.00140). However, to better clarify the title that in fact is ambiguous, as also noted by the Reviewer, we propose the new title: “What can mushroom proteins teach us about lipid rafts?”
Query #2:
It would be good if the authors could briefly discuss what is the natural function of these proteins in mushrooms.
Response from the Authors:
The partial answers of these comments already lie within the text (bolded in yellow). As cited in the article, it was found for ostreolysin A that it is expressed during the formation of primordia and fruiting bodies of the oyster mushroom, and therefore it was believed to have a role in mushroom fruiting (lines 135 – 139). Furthermore, as discussed in the text, aegerolysins have another high-affinity membrane receptor besides the mixture of sphingomyelin and cholesterol. This is the invertebrate-specific sphingolipid ceramide phosphoethanolamine. Thus, fungal aegerolysins together with their MACPF-partnering proteins might be responsible for mushroom defence mechanism against some pest predators as discussed in lines 207 – 218. Since this paragraph might be misleading we introduced a new sentence in the text (lines 218 -221, bolded in yellow).
Query #3:
I do not quite agree with reasoning behind criticism of older methods to investigate lipid rafts. At times, this reasoning simply represents Heisenberg’s Uncertainty Principle (to investigate the object you need to disturb it – apply detergents, or insert a fluorescent probe etc), which applies to any method and is not a valid point of criticism of any particular method. The methods mentioned do have many limitations, it’s just not this.
Response from the Authors:
We agree with the Reviewer’s #2 criticism. All the lipid rafts probes and tools have advantages and disadvantages. We have included in the text also the main disadvantage of mushroom lipid raft-binding proteins (lines 105 -108).
Query #4:
Lines 170-172: a straight forward explanation is that OlyA6-mCherry does not bind to lipid rafts, but to a different domain, isn’t it? This needs to be discussed.
Response from the Authors:
Lipid microdomains are highly heterogeneous and there is a great need for further research that would allow to find the answers for yet unanswered questions. Among the most intriguing questions that still remain unanswered are: how do these domains behave, what is their exact composition and how they interact whit other domains in the surroundings. According to Endapally et al., (Cell 2019, 176, 1040-1053.e17, DOI: 10.1016/j.cell.2018.12.042), ostreolysin A6 binds exclusively to the combination of sphingomyelin and cholesterol, meaning that it binds to lipid rafts in mammalian cell membranes. Majority of lipid raft researchers also agree that lipid rafts are highly heterogeneous (Sezgin et al. Nat Rev Mol Cell Biol 18, 361–374 (2017); Mishra et al. J Neurochem. 2007; 103 Suppl 1:135-42) and it might be that specific protein-based lipid raft markers sense distinctive raft-like membrane microdomains which are yet to be discovered. For example, as cited in the text, cholera toxin subunit B uses ganglioside GM1 as its raft-residing target, and consequently oligomerizes, which can lead to the formation of sub group of lipid raft platforms. Therefore, in order to correctly define, observe and interparty the function and behaviour of these domains in living vertebrate cells all the data obtained with lipid-binding probes have to be critically evaluated, which was also one of the main aims of the present review. Since ostreolysin A6 binds to sphingomyelin/cholesterol-rich membrane domains, the sentence in lines 175-176 (bolded in yellow) was revised as: “Co-localization studies showed that CT-B labels different domains compared to OlyA6, which confirmed that raft domains are highly heterogeneous (Figure 2).”
Query #5:
Minor: my understanding is that the target audience of this review are experts in membranes, not experts in mushrooms. I would recommend including some brief explanations of botanical terms, it will make reading stumble-free.
Response from the Authors:
We are not sure to which botanical terms does the Reviewer #2 refers. Perhaps she/he is referring to the terms that describe different developmental stages of mushrooms? In any case, we have now explained these terms in the revised version of the manuscript (lines 135-139, bolded in yellow): ”It is expressed during the formation of primordia (an organ or tissue in its earliest recognizable stage of development) and fruiting bodies (a macroscopic reproductive structure produced by some fungi) of the oyster mushroom, and therefore it was believed to have a role in mushroom fruiting [41, 44].«
